# Placing *Ureaplasma* within the Context of Bronchopulmonary Dysplasia Endotypes and Phenotypes

**DOI:** 10.3390/children10020256

**Published:** 2023-01-31

**Authors:** Karen Van Mechelen, Elke van Westering-Kroon, Matthias Hütten, Ludo Mahieu, Eduardo Villamor

**Affiliations:** 1Department of Pediatrics, Maastricht University Medical Center, School for Oncology and Reproduction (GROW), Maastricht University, 6229 HX Maastricht, The Netherlands; 2Department of Neonatology, Antwerp University Hospital, 2650 Edegem, Belgium

**Keywords:** bronchopulmonary dysplasia, BPD, *Ureaplasma*, macrolides, endotypes, phenotypes

## Abstract

Different pathophysiological pathways (endotypes), leading to very preterm birth may result in distinct clinical phenotypes of bronchopulmonary dysplasia (BPD). *Ureaplasma* is a unique player in the pathogenesis of BPD. The interaction between factors inherent to *Ureaplasma* (virulence, bacterial load, duration of exposure), and to the host (immune response, infection clearance, degree of prematurity, respiratory support, concomitant infections) may contribute to BPD development in a variable manner. The data reviewed herein support the hypothesis that *Ureaplasma*, as a representative of the infectious/inflammatory endotype, may produce pulmonary damage predominantly in parenchyma, interstitium, and small airways. In contrast, *Ureaplasma* may have a very limited role in the pathogenesis of the vascular phenotype of BPD. In addition, if *Ureaplasma* is a key factor in BPD pathogenesis, its eradication by macrolides should prevent BPD. However, various meta-analyses do not show consistent evidence that this is the case. The limitations of current definitions and classifications of BPD, based on respiratory support needs instead of pathophysiology and phenotypes, may explain this and other failures in strategies aimed to prevent BPD. The precise mechanisms through which *Ureaplasma* infection leads to altered lung development and how these pathways can result in different BPD phenotypes warrant further investigation.

## 1. Introduction

More than 50 years after its initial description [1], bronchopulmonary dysplasia (BPD) continues to be one of the most significant complications of prematurity and still poses clinical and public health challenges [2,3,4,5,6,7]. Northway et al. coined the term BPD to describe a chronic pulmonary condition observed in relatively late preterm infants (32–34 weeks) with respiratory distress syndrome and treated with mechanical ventilation and high oxygen concentrations [1]. However, the population of infants currently diagnosed with BPD differs significantly from the original description. What we now call BPD is the consequence of the exposure to extrauterine life of lungs in the late canalicular or early saccular stage of development (i.e., below 28 weeks of gestation) [3,4]. This results in alterations in both alveolar and pulmonary vascular development that can also occur without exposure to mechanical ventilation or high oxygen concentrations. Therefore, the contemporary view of BPD is that of a heterogeneous condition in which the combination of genetic background, antenatal exposures, extreme prematurity, and early postnatal exposures lead to different patterns of injury and repair [3,4]. Moreover, there is a growing awareness that distinct intrauterine stresses may result in different clinical phenotypes of BPD beyond the traditional categorization of mild/moderate/severe [8,9,10].

The term endotype refers to “a subtype of a condition, which is defined by a distinct functional or pathophysiological mechanism” [11]. The different pathophysiological pathways that trigger extremely preterm birth and concurrently induce alterations in fetal lung development can be grouped into two endotypes: infectious/inflammatory and placental dysfunction [8,9,10,12]. In addition, a growing body of evidence indicates that airways, alveoli, lung parenchyma, and pulmonary vasculature are variably affected within each infant, which leads to different clinical phenotypes of BPD. The following seven phenotypes of BPD have been proposed: parenchymal, peripheral airway, central airway, interstitial, congestive, vascular, and mixed phenotype [8,13,14,15,16,17,18]. These phenotypes may have a partial or complete origin in prematurity endotypes [8,9,10]. Therefore, endotyping and phenotyping BPD will increase our understanding of the condition and facilitate the design of more targeted preventive and therapeutic approaches [3,8,9,10,13].

*Ureaplasma* is recognized as a major pathogen in prenatal infection triggering preterm birth and is frequently claimed to be involved in the pathogenesis of BPD [19,20,21,22]. As *Ureaplasma* is the most frequently isolated bacteria in the female genital tract [23,24], fetal or neonatal colonization may occur prenatally due to ascending intrauterine or placental infection, during birth by passage through the birth canal, or postnatal [19,24,25,26,27,28]. The incidence of *Ureaplasma* colonization in preterm neonates ranges from 15 to 36% [19,29,30,31,32,33,34,35,36]. The lower the gestational age (GA) or birth weight (BW), the higher the colonization rate [19,27]. 

Numerous preclinical and clinical studies, reviews, and meta-analyses have addressed the association between *Ureaplasma* and BPD but the results and conclusions are often contradictory [22,27,37,38,39]. Some of these contradictions may stem from considering BPD as a homogeneous entity rather than accounting for the possibility that different endotypes result in different BPD phenotypes. Therefore, we still do not fully understand whether *Ureaplasma* plays a major role in BPD development, is just a cofactor, or is just an innocent bystander. Our aim in this narrative review is to summarize and update information on the role of *Ureaplasma* in the pathogenesis of BPD and to frame it within the context of endotyping and phenotyping BPD.

## 2. Virulence of and Immune Response to *Ureaplasma*


*Ureaplasma* are spherical or coccobacillus-shaped bacteria that belong to the Mycoplasmataceae family. *Ureaplasma* bacteria look much like Mycoplasma but they hydrolyze urea to generate adenosine triphosphate, rather than arginine or glutamine. Furthermore, *Ureaplasma* lack a cell wall, are not visible on Gram staining and are resistant to β-lactam antibiotics. There are two separate species, namely *Ureaplasma parvum* and *Ureaplasma urealyticum,* with 14 serovars that can infect humans [40]. *Ureaplasma* bind to mucosal surfaces and are typically found in the mouth, upper respiratory and urogenital tract [41]. In women, they are the most frequently isolated commensal in the lower genital tract [23,41].

*Ureaplasma* can be virulent through different mechanisms, including multiple banded antigen (MBA), production of ammonia by urease, immunoglobulin A1 protease, lack of cell wall, and phospholipases [42,43]. These virulence factors evade the host immune response and can cause tissue damage [42]. The immune system then reacts producing cytokines such as tumor necrosis factor α, interleukin (IL)-6, IL-8, or IL-10 [44], activating the nuclear factor kappa B (NFkB) pathway via toll-like receptors (TLR)-1, TLR-2, and TLR-6 [45], stimulating macrophage activity through TLR-2 and TLR-4 [46], and increasing an immunoglobulin response [47]. At least part of this immune response has a protective character [47]. Another protective mechanism shown in in vitro models is surfactant protein A, which clears *Ureaplasma* in mouse lungs [48]. However, since surfactant production is inversely proportional to GA, extremely preterm newborns partially lack this protective mechanism. 

## 3. *Ureaplasma* and Pregnancy 

*Ureaplasma* are the most frequent commensal in the female lower genital tract and are also capable of causing ascending asymptomatic infections of the upper genital tract [23,24,49]. Ascending *Ureaplasma* in pregnant women can induce an inflammatory response in choriodecidua and amnion, resulting in cytokines and prostaglandins production, and stimulation of uterine contractions [50]. These reactions are associated with chorioamnionitis, premature rupture of membranes (PROM), preterm birth, and abortion [22,24,49,51,52,53,54,55]. Interestingly, only a small group of women with *Ureaplasma* colonization/infection develop these complications [49,51,52,53,54]. As reviewed by Sweeny et al., among the factors claimed to explain these differences are the virulence of the infecting serovar, the bacterial load, genetic background/ethnicity, and variations in the host immune response [49,56,57].

One of the main arguments for questioning the role of *Ureaplasma* as a major pathogen in chorioamnionitis is the frequent polymicrobial nature of the condition [49,58,59,60]. This seems to be particularly true in the case of chorioamnionitis below 32 weeks of gestation in which the presence of two or more microorganisms has been demonstrated in more than 60% of cases [49]. Of note, *Ureaplasma* is almost always one of these microorganisms and as GA advances, the proportion of cases in which it is the only bacteria identified increases [49]. 

## 4. *Ureaplasma* and the Developing Lung: Evidence from Preclinical Studies

The effects of *Ureaplasma* infection in the fetal and neonatal lung have been studied mainly in sheep and non-human primates [21]. Intraamniotic injection of *Ureaplasma parvum* in early gestation sheep (55 of 150 days) resulted in chorioamnionitis and lung colonization accompanied by a modest increase in neutrophils, monocytes, IL-1β and IL-8 [61]. Similarly, exposure to *Ureaplasma parvum* on day 80 of pregnancy in sheep also results in chorioamnionitis and modest fetal lung inflammation after 6 weeks [62]. This lung inflammation persists for 9-10 weeks [62]. Moreover, exposure to *Ureaplasma* in sheep during late-gestation (110 to 121 days) induced a mild acute inflammatory response with temporary increased neutrophils in preterm lungs [63]. In mice, a self-limiting lung inflammatory response was demonstrated after exposure to *Ureaplasma parvum* [64]. In contrast, non-human primates shortly exposed to *Ureaplasma parvum* during mid-gestation have strong lung inflammation [65]. Fetal baboons infected with *Ureaplasma* were seen to have increased cytokine expression in their lung fluid, causing severe bronchiolitis, and interstitial pneumonitis [66]. In addition, antenatal exposure to *Ureaplasma* can influence the hosts’ reaction to postnatal stimuli. For example, antenatal *Ureaplasma* exposure in sheep alters the innate immune system making the immature lung more vulnerable to inflammatory stimuli after birth [63]. Furthermore, studies in mice and baboons demonstrated an increased response to postnatal stimuli such as oxygen and mechanical ventilation [64,66]. It seems that the host response differs between non-primates and non-human primates, and influences lung inflammation and therefore lung morphology changes [21].

There is some evidence for long-term alveolar or vascular morphology changes in non-primate models [61,64]. Studies demonstrated an increased lung maturation, with increased surfactant and lung volumes, in sheep that had long-term *Ureaplasma* exposure [61], and increased surfactant but no change in lung volumes in sheep that had mid-long-term *Ureaplasma* exposure [62]. In sheep with short-term *Ureaplasma* exposure, there were decreased elastic foci with increased smooth muscle around bronchioles, and pulmonary vasculature [63]. This decreased elastin results in lung simplification [67]. These lung changes in non-primates were independent of the dose of *Ureaplasma* exposure or serovar [61]. In contrast, non-human primates exposed to *Ureaplasma* had more severe lung alternations compared to sheep exposed to *Ureaplasma* [21]. In non-human primate models, it seems that the acute lung inflammatory response can partially resolve but the structural lung changes such as epithelial hyperplasia, thickened alveolar walls, fibrosis and bronchiolitis remain [65,66,68,69]. Nevertheless, baboons that could clear their *Ureaplasma* infection had lower respiratory requirements on mechanical ventilation with improved lung function, in contrast to those which could not clear the infection [66]. 

The results of these preclinical studies on *Ureaplasma* are in contrast with the results of studies on chorioamnionitis caused by the bacterial endotoxin lipopolysaccharide (LPS). Animals exposed to chorioamnionitis due to injection of LPS developed a stronger inflammatory response than animals exposed to *Ureaplasma*. Sheep exposed to LPS had higher concentrations of inflammatory cells such as neutrophils, monocytes, IL-1β [62,70,71], and macrophages [72] in their lungs in comparison with sheep exposed to *Ureaplasma*. Furthermore, the inflammatory response from TLR4 receptors through LPS is stronger than the inflammatory response from TLR1, 2, and 6 through *Ureaplasma* [45,73]. Nevertheless, common underlying mechanisms are suggested. For example, pulmonary leukocyte-mediated lung inflammation stimulates lung maturation in sheep exposed to LPS and in sheep exposed to *Ureaplasma* [74]. Despite the more severe inflammatory endotype caused by LPS in the lungs of sheep, there was no sustained change in lung morphology such as decreased alveolarization [75]. In contrast, in LPS-exposed sheep the inflammation inhibited vascular endothelial growth factor (VEGF), endothelial nitric oxide synthase (eNOS), platelet endothelial cell adhesion molecule-1, and Tie-2 protein, which led to vascular remodeling in small pulmonary arteries causing pulmonary hypertension [75]. 

In summary, preclinical studies demonstrated that *Ureaplasma* induces a mild inflammatory response in fetal lungs. Despite the milder inflammatory endotype, in comparison with the stronger inflammatory response caused by acute chorioamnionitis due to other etiologies (such as LPS), *Ureaplasma* causes more structural lung changes. Interestingly, a longer fetal *Ureaplasma* exposure induces lung maturation without sustained structural changes, whereas a shorter fetal *Ureaplasma* exposure causes more sustained structural changes [61,63]. This underlines the unique character of *Ureaplasma*.

## 5. *Ureaplasma*: Causative Agent, Cofactor, or Innocent Bystander in BPD Pathogenesis?

The positive association between maternal colonization with *Ureaplasma* and BPD (OR 2.4; 95% CI 1.7–3.3) has been shown in a very recent meta-analysis [22]. To the best of our knowledge, four meta-analyses attempted to synthesize the evidence of the association between pulmonary colonization with *Ureaplasma* and BPD [27,37,38,39]. The results of these meta-analyses are summarized in Table 1. Overall, the data suggest an increased risk of developing BPD in infants colonized with *Ureaplasma* but there are some limitations of the meta-analyses that should be taken into account. First, they had moderate/high degrees of heterogeneity. Part of this heterogeneity may be explained because the effect size decreased as the included studies became more recent. Of note, Zhang’s meta-analysis [37] only included studies published since 1995 and is the only one not reporting a significant association (Table 1). In addition, many of the studies included in the meta-analyses had very small sample sizes so there is a risk of so-called "small study effects" [76]. That is, smaller studies sometimes show different, often larger, treatment effects than larger studies [76]. Although there is no intrinsic inaccuracy in small studies with large effect sizes, selective publication of the results of smaller studies is more common than that of larger studies and this can seriously affect meta-analysis results [76]. Finally, the meta-analyses included two definitions of BPD: based on oxygen requirement at 28 days postnatal (hereinafter referred to as BPD28) and at 36 weeks postmenstrual age (hereinafter referred to as BPD36). The association of *Ureaplasma* colonization with BPD36 is weaker than the association with BPD28 (Table 1). Interestingly, the same is observed for the association between chorioamnionitis and BPD [77]. Neonatologists have long argued about the inadequacy of the diagnostic criteria for BPD and particularly about the fact that they are based on components of care (oxygen or respiratory support needs) and not on pathophysiology [3,4]. In any case, oxygen requirement at postnatal day 28 is probably only a proxy for pulmonary immaturity, while BPD36 is more reflective, albeit still imperfectly, of pulmonary damage or aberrant development [3,4]. 

Irrespective of the quality of the evidence on the association between *Ureaplasma* colonization and BPD, association does not mean causation. The possible role of *Ureaplasma* as a causative agent of BPD may depend on the interaction of some factors, not yet fully understood, between the bacteria and the host. Thus, it has been suggested that the type of serovar is a determining pathogenic factor for BPD [78] but this has not been confirmed by other studies [55]. In addition, the duration of colonization/infection with *Ureaplasma*, bacterial load, and longitudinal changes in the airway microbiome may be determining factors in the development of BPD [61,63,79,80,81]. Paradoxically, the data from preclinical studies suggest that longer fetal *Ureaplasma* exposure may accelerate lung maturation without inducing structural changes, whereas a shorter fetal exposure may cause more sustained structural changes [61,63]. *Ureaplasma* and *Staphylococcus* are the predominant genera of the early airway microbiome of very preterm infants [80,81]. In the first days after birth, infants with more severe BPD had a greater initial relative abundance of *Ureaplasma* and acquired less *Staphylococcus* [80,81]. 

Finally, if *Ureaplasma* is an important causative agent in BPD development, eradication by macrolides should significantly reduce BPD. However, this reduction has not been consistently demonstrated by meta-analyses [82,83,84]. The failure of macrolide treatment to reduce BPD might be due to some proportion of preterm infants requiring oxygen at 36 weeks postmenstrual age but having a BPD phenotype that is not associated with the infectious/inflammatory endotype. This phenotype would be therefore indifferent to antibiotic therapy. The relationship between endotype and phenotype is discussed in the following paragraphs.

## 6. Linking *Ureaplasma* Endotype with BPD Phenotypes

There is overwhelming evidence of the relevance of *Ureaplasma* as a causative agent of chorioamnionitis [22] and chorioamnionitis is the main proxy of the infectious/inflammatory endotype of prematurity [8,9,10]. Previous meta-analyses have demonstrated a robust association between chorioamnionitis and risk of developing BPD28 and BPD36, but not severe BPD (defined as the need for more than 30% oxygen or mechanical ventilation at 36 weeks postmenstrual age) [77,85]. However, the effect of chorioamnionitis was strongly related to the fact that infants in the control group had significantly higher GAs than those in the chorioamnionitis group [77]. It is a well-known fact that the incidence of chorioamnionitis increases as GA decreases and therefore the majority of extremely preterm newborns belong to the infectious/inflammatory endotype [86,87]. In addition, the presence of funisitis did not add an additional risk of BPD when compared to chorioamnionitis in the absence of funisitis [77]. Funisitis is considered the histological equivalent of the fetal inflammatory response syndrome, whereas chorioamnionitis represents a maternal inflammatory response [86]. Taken together, the meta-analysis data suggest that a significant part of the association between chorioamnionitis and BPD is mediated by the role of chorioamnionitis as a trigger of prematurity more than by the alteration that it induces in lung development [77]. It is remarkable that infants colonized by *Ureaplasma* had a significantly lower GA than controls and Lowe et al. demonstrated by meta-regression that there was a correlation between this difference in GA and the effect size of the association with BPD [27]. In summary, the induction of a higher degree of prematurity seems to be an inherent feature of the inflammatory/infectious endotype when compared to the placental dysfunction endotype. This does not preclude an additional, endotype-specific, pulmonary damage mediated by the inflammatory cascade. 

As mentioned in the introduction, up to seven different clinical phenotypes of BPD can be distinguished but there is still a great lack of consensus on how to diagnose these phenotypes [8,9,10]. The one that may be best defined is the vascular phenotype of BPD, which is characterized by the concomitant presence of pulmonary hypertension [15,16,17,18,88,89,90]. A great effort has been made in recent years to reach a consensus on how and when to assess BPD-associated pulmonary hypertension [91]. Nevertheless, the vascular phenotype of BPD is most frequently associated with the placental dysfunction endotype of prematurity [8,9,10]. This is because the imbalance of pro- and anti-angiogenic mediators that takes place in the dysfunctional placenta also affects pulmonary vascular development [8,9,10]. In contrast, the damage caused by the infectious/inflammatory endotype seems to affect more preferentially the development of alveoli and pulmonary interstitium [8,9,10]. In addition, there is growing evidence that pulmonary infection/inflammation may increase the sensitivity of the lung to postnatal stresses such as mechanical ventilation [92,93].

Autopsy studies in *Ureaplasma*-infected preterm lungs showed an increased concentration of alveolar macrophages, myofibroblast proliferation, abnormal septation, interstitial fibrosis, and disordered elastic fibers in the distal air spaces [94,95]. These findings are similar to those of the interstitial BPD phenotype, which is characterized by an increase in fibrotic tissue and a widening of the interstitial spaces [8,96]. Although it is difficult to derive specific information from chest radiographs, it is noteworthy that *Ureaplasma* infection/colonization in extremely preterm infants has been associated with cystic-emphysematous changes [97,98]. This may be suggestive of a parenchymal BPD phenotype [8]. Finally, Kitajima et al. reported that intrauterine *Ureaplasma* infection in extremely preterm infants was associated with signs of moderate to severe small airway obstruction in lung function tests at school age [97]. These findings are compatible with the peripheral airway BPD phenotype [8]. Taken together, all these data suggest that *Ureaplasma* may be associated with a mixed phenotype in which the pulmonary parenchyma and small airways would be more affected while the pulmonary vasculature would be less involved. It could be speculated that this particular form of BPD might be more susceptible to treatment with corticosteroids and/or β2-agonists. In contrast, screening for pulmonary hypertension and treatment with pulmonary vasodilators would be preferentially indicated in the vascular phenotype of BPD [8,9,10]. This phenotype is more linked to the placental dysfunction endotype than to the infectious/inflammatory endotype [8,9,10].

## 7. BPD and Eradication of *Ureaplasma*

### 7.1. Prenatal

Macrolides are the antibiotics of choice for eradicating *Ureaplasma* in neonates and pregnant women because they are not, unlike fluoroquinolones and tetracyclines, teratogenic. Maternal macrolides can reduce vertical *Ureaplasma* transmission to the fetus and prevent the development of the inflammatory cascade. The literature is contradictory as to whether or not maternal macrolides prevent adverse pregnancy outcomes [99]. It is difficult to compare studies, as there are large differences between particular macrolides, dosing regimens, and routes of administration. In primate models, intramuscular or intra-amniotic erythromycin seems insufficient to eradicate *Ureaplasma* [100] because erythromycin does not pass the placenta. In contrast, solithromycin and azithromycin seem sufficient either through intravenous or by combining intravenous and intra-amniotic, administration [101,102,103]. Others, however, state that a combination of maternal (intravenous or intramuscular) and intra-amniotic administration is necessary to achieve therapeutic concentrations in both maternal and amniotic compartments [104]. Research also noted that in women with preterm pre-labor rupture of membranes (PPROM), antibiotics prolong pregnancy, decrease chorioamnionitis, as well as neonatal infection, and oxygen therapy [101,105,106]. In contrast, other studies demonstrate no or partial benefits [107,108]. Therefore, the efficacy of maternal therapy for *Ureaplasma* remains uncertain.

Furthermore, it is important to notice the possible side effects of macrolides. A meta-analysis of 2020 identified a weak association between the use of macrolides in early pregnancy and congenital malformations of the musculoskeletal and digestive systems [109]. The developing immune and neurological systems can already be influenced by one course of antibiotics which alter the microbiome of the neonate and mother [110,111]. In addition, intrapartum antibiotics increase antibiotic resistance in neonates [112].

### 7.2. Postnatal

Three meta-analyses about macrolide treatment in *Ureaplasma*-positive neonates reported variable results [82,83,84]. A Cochrane review of 2003 included two small RCTs and found no significant reduction in BPD risk when erythromycin was used in intubated preterm neonates [84]. Similar findings were reported by Nair et al. that included six RCTs [83]. This meta-analysis found no significant reduction in BPD36 (RR 0.64; 95% CI 0.31–1.31) or the composite outcome BPD36 or death (RR 0.41; 95% CI 0.05–3.13) when clarithromycin, azithromycin or erythromycin was used in *Ureaplasma*-positive neonates [83]. They did, however, report a significant reduction in BPD36 and BPD36 or death with prophylactic azithromycin irrespective of *Ureaplasma* status [83]. A possible explanation for this reduction might be that azithromycin and clarithromycin achieve better drug concentrations in the lungs than erythromycin [113]. Furthermore, the used dose of the macrolide is important to determine *Ureaplasma* clearance [114]. A limitation of this meta-analysis is that they combined studies having important differences in population, the drug used, dosages, time of initiation, duration of treatment, and method of *Ureaplasma* detection. The third meta-analysis by Razak and Alshehri, published in 2021, only evaluated azithromycin therapy [82]. Five RCTs were included (four including *Ureaplasma*-positive neonates). The meta-analysis demonstrated that *Ureaplasma*-positive neonates treated with azitromycin had less BPD36 or death (RR 0.83; 95% CI 0.70–0.99), and a trend toward less BPD36 (RR 0.83; 95% CI 0.66–1.03) [82]. In addition, there was a reduction in supplemental oxygen days, in all neonates irrespective of *Ureaplasma* status [82]. These findings suggest that azithromycin can effectively attenuate the pathogenic effect of *Ureaplasma* on BPD. Azithromycin has high potency against *Ureaplasma* [32] and has anti-inflammatory capacities by reducing IL-2 and IL-8 levels [31] which is an important pathway in BPD development. Nevertheless, the quality of evidence was low due to the risk of bias among the RCTs and the imprecision of the effect estimate [82]. 

The number needed to treat with macrolides to prevent one case of BPD was 11 in the meta-analysis of Razak and Alsheri [82] and 10 in the meta-analysis of Nair et al. [83]. Therefore, it is very important to select the infants who will benefit the most from treatment with macrolides. Extremely preterm neonates and neonates who cannot be extubated, are considered at high risk for BPD. Poets et al. suggests testing tracheal aspirates of extremely preterm neonates requiring mechanical ventilation for *Ureaplasma*, and consider intravenous azithromycin in those who tested positive [115]. Nevertheless, today, an increasing number of infants with BPD were never intubated due to advances in non-invasive ventilation. 

It is important to bear in mind that comparing individual studies is difficult because there is considerable heterogeneity due to the use of different macrolide regimes (drug, dosing, duration), inclusion criteria, baseline characteristics, ventilation techniques, BPD definitions, sample sizes, and statistical power. To conclude, the safety and effectiveness of macrolides to eradicate *Ureaplasma* remains uncertain. Although there is some evidence that the macrolide azithromycin has an overall anti-inflammatory effect [31] and reduces BPD in *Ureaplasma*-positive neonates [82], the use of macrolides is not recommended because the current level of evidence is weak. The ongoing AZTEC (azithromycin therapy for chronic lung disease of prematurity) aims to determine if a 10-day course of intravenous azithromycin improves rates of survival without BPD36 when compared with placebo in a population of infants with GA <30 weeks [116].

## 8. Conclusions and Suggestions for Future Research

*Ureaplasma* is a unique player in the pathogenesis of BPD. The interaction between factors inherent to *Ureaplasma,* (virulence, bacterial load, duration of exposure), and to the host (immune response, infection clearance, degree of prematurity, respiratory therapy needs, concomitant infections) contribute, in a manner yet to be fully determined, to BPD development. The data reviewed herein support the hypothesis that *Ureaplasma*, as a representative of the infectious/inflammatory endotype, produces pulmonary damage predominantly in parenchyma, interstitium, and small airways. However, the precise mechanisms through which *Ureaplasma* infection leads to altered lung development and how these pathways can result in different BPD phenotypes require ongoing exploration. 

In addition to *Ureaplasma*, very preterm infants are exposed to multiple potent antenatal and postnatal inflammatory modulators that may modify pulmonary injury and repair pathways. This results in a marked variability in clinical manifestations and severity of BPD. Current definitions of BPD are based on components of care, such as the need for oxygen and/or respiratory support, and do not capture this variability [3,4]. Although the persistent requirement of oxygen or respiratory support at a given age may be a clinical predictor of respiratory outcome, it provides little information about BPD phenotypes or therapeutic interventions that may benefit individual patients. A classification of BPD based on endotypes and phenotypes requires the development of biomarkers targeting *Ureaplasma* infection as well as other pathogenic pathways [3,4]. 

A challenge in future research is diagnosing *Ureaplasma* infection. The gold standard for diagnosing *Ureaplasma* is bacterial culture in a specialized media. *Ureaplasma* bacteria lack a cell wall and are therefore highly sensitive to desiccation and heat. That means any test samples must be immediately transported to the laboratory, and culturing *Ureaplasma* remains difficult [117]. To overcome these difficulties, polymerase chain reaction (PCR) can be helpful. In this respect, some suggest that PCR and culture are almost equally effective [118,119], yet others suggest that PCR is superior [120]. PCR tests have the advantage of allowing the identification of *Ureaplasma* species, the quantification of bacterial load, and the evaluation of specimens overgrown by other bacteria using the culture method. As discussed above, all of these factors are relevant to the pathogenic effects of *Ureaplasma* [78,79,119]. Nevertheless, a disadvantage of PCR is that it is not possible to test antimicrobial susceptibility, in contrast to culturing. A disadvantage of both tests is that neither can distinguish between carrier status and infection. 

Finally, the evidence for treating *Ureaplasma,* and therefore decreasing BPD, remains inconclusive. Future studies are necessary to find the optimal approach for *Ureaplasma* in pregnant women and their offspring. The ideal drug, dosage regimen, timing, and route of administration remains a matter of debate. Furthermore, it remains unclear whether treating the infection/colonization with *Ureaplasma* effectively decreases lung inflammation, reduces BPD rate, and reduces long-term respiratory and neurological morbidity. 

## Figures and Tables

**Table 1 children-10-00256-t001:** Summary of meta-analyses on the association between neonatal pulmonary colonization with *Ureaplasma* and bronchopulmonary dysplasia.

Meta-Analysis	BPD28			BPD36		
	Effect Size	K	*p*-Value Heterogeneity	Effect Size	K	*p*-Value Heterogeneity
Wang et al. 1995 [39]	RR 1.8 (95% CI 1.5–2.0)	17	0.09			
Schelonka et al. 2005 [38]	OR 2.8 (95% CI 2.3–3.5)	23	<0.01	OR 1.6 (95% CI 1.1–2.3)	8	<0.01
Lowe et al. 2014 [27]	OR 3.0 (95% CI 2.4–3.8)	31	0.32	OR 2.22 (95% CI 1.4–3.5)	17	<0.001
Zhang et al. 2014 [37]				OR 1.1 (95% CI 0.7–1.7)	11	<0.001

BPD28: Bronchopulmonary dysplasia defined by oxygen requirement at 28 days postnatal; BPD36: Bronchopulmonary dysplasia defined by oxygen requirement at 36 weeks postmenstrual age; CI: confidence interval; K: number of studies included in the meta-analysis; OR: odds ratio; RR: risk ratio.

## Data Availability

Not applicable.

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
