# Peer review of "Placing Ureaplasma within the Context of Bronchopulmonary Dysplasia Endotypes and Phenotypes"

_children, 2023, doi:10.3390/children10020256_

Round 1
Reviewer 1 Report
This is a review article. The authors conclude that Ureaplasma is a unique player in the pathogenesis of bronchopulmonary dysplasia (BPD). The interaction between factors inherent to Ureaplasma and to the host contribute to BPD. Besides Ureaplasma, very preterm infants are exposed to multiple potent antenatal and postnatal inflammatory modulators that may modify pulmonary injury and repair pathways.
This is an interesting and valuable review article.
Author Response
Thank you very much for your kind words and positive evaluation of our manuscript.
Reviewer 2 Report
Congratulations on the job.
In this work, the authors put Ureaplasma colonization/infection in the preterm newborn in context, based on the endotypes of prematurity and the described phenotypes of BPD. The approach is extremely interesting and quite novel. The literature review is appropriate and comprehensive. The work is very well written and organized. Excellent presentation.
Minor comments.
Some references seem to be incomplete. For example:
Reference 10, lines 429-430 (pages?): Parsons A, Netsanet A, Seedorf G, Abman SH, Taglauer ES. Understanding the role of placental pathophysiology in the development of bronchopulmonary dysplasia. Am J Physiol Lung Cell Mol Physiol. 2022 Dec 1;323(6):L651-L658. doi: 10.1152/ajplung.00204.2022. Epub 2022 Oct 11. PMID: 36219136; PMCID: PMC9722259.
Ref. 15 (vol, pages?).
Ref. 18.
And some other…
Please, review and complete them.
Please verify also the correct abbreviation of the names of the journals. For instance:
Ref 88: Clinics in perinatology: Clin Perinatol
Ref 89: Journal of Perinatology: J Perinatol
Ref 90: American Journal of Physiology-Lung Cellular and Molecular Physiology: Am J Physiol Lung Cell Mol Physiol.
Ref 91: The Journal of Pediatrics: J Pediatr.
Etc.
Also, be consistent with punctuation signs in the abbreviations.
Typos.
Lines 306-7: “A meta-analysis of 2002”, should be “A meta-analysis of 2020”.
Line 320: If appropriate, consider “BPD36 and BPD36 or death” instead of “BPD and BPD or death”, for consistency.
Author Response
Thank you very much for your kind words and positive evaluation of our manuscript.
We edited the references and completed them.
Typos.
Lines 306-7: “A meta-analysis of 2002”, should be “A meta-analysis of 2020”.
This is indeed incorrect, we changed 2002 into 2020.
Line 320: If appropriate, consider “BPD36 and BPD36 or death” instead of “BPD and BPD or death”, for consistency.
We changed “BPD and BPD or death” into “BPD36 and BPD36 or death”.
Reviewer 3 Report
Placing Ureaplasma within the Context of Bronchopulmonary Dysplasia Endotypes and Phenotypes by Karen Van Mechelen, et al.
The role of Ureaplasma in development of BPD is analyzed with a complete revision and useful suggestions for caregivers.
Author Response

(The authors gave the same response as above.)
